# On the applicability of linear wave theories to simulations on the mid-latitude $\beta$ -plane

Itamar Yacoby<sup>1,2</sup>, Hezi Gildor<sup>2</sup>, and Nathan Paldor<sup>2</sup>

<sup>1</sup>Present Affiliation: Department of Geophysics, Porter School of the Environment and Earth Sciences, Tel Aviv University, Tel Aviv, Israel

<sup>2</sup>Fredy and Nadine Herrmann Institute of Earth Sciences, Edmond J. Safra Campus, Givat Ram, the Hebrew University of Jerusalem, Jerusalem, Israel

Correspondence: Nathan Paldor (nathan.paldor@mail.huji.ac.il)

Abstract. The applicability of one-dimensional (zonally invariant) harmonic and trapped wave theories for Inertia-Gravity waves to simulations on the mid-latitude  $\beta$ -plane is examined by comparing the analytical estimates in the geostrophic adjustment and Ekman adjustment problems with numerical simulations of the linearized rotating shallow water equations. The spatial average of the absolute differences between the theoretical solutions and the simulations,  $\epsilon(t)$ , is calculated for values of the domain's north-south extent, L, ranging from L=4 to L=60 (where L is measured in units of the deformation radius). The comparisons show that: (i) Though  $\epsilon$  oscillates with time, its low-pass filter,  $\epsilon^{LP}(t)$ , increases with time. (ii) In small domains,  $\epsilon^{LP}(t)$  in harmonic theory is significantly smaller than in trapped wave theory, while the opposite occurs in large domains. (iii) The accuracy of the harmonic wave theory decreases with L for 0 < L < 20, while for L > 20 the trend changes with time. (iv) The accuracy of the trapped wave theory increases with L in the geostrophic adjustment problem, its best accuracy is obtained when  $L \approx 30$ . (v) There is a range of L and t values for which no theory provides reasonable approximations, and this range is wider in the Ekman adjustment problem than in the geostrophic adjustment problem. Non-linear simulations of a multilayered stratified ocean show that internal inertia-gravity waves exhibit the same characteristics as the wave solutions of the linearized rotating shallow water equations in a single layer ocean.

Copyright statement. TEXT

#### 15 1 Introduction

The Rotating Shallow Water Equations (RSWE, hereafter) provide a fundamental description of the dynamics of an incompressible fluid in a thin layer in the presence of rotation. This framework is applicable when the horizontal scale of the fluid motion is much larger than the layer thickness. The linear waves of the RSWE include three wave types: Kelvin waves, Inertia-Gravity waves (also known as Poincaré waves) and Planetary waves (also known as Rossby waves). Mid-latitude (coastal) Kelvin waves occur in the presence of an ocean boundary, while all three wave types are generated in response to atmospheric forcing, such as wind stress, or due to local perturbations in the ocean's velocity or surface height. These waves are tradition-

ally classified into two main categories based on their frequencies. The first category comprises the high-frequency Kelvin and Inertia-Gravity waves, which are rotationally modified gravity waves. The second category includes the low-frequency Planetary waves, which originate as perturbations respond to the latitudinal variation of the Coriolis parameter [see, e.g., Gill (1982), Pedlosky (1987), Cushman-Roisin and Beckers (2011), and Vallis (2017)].

In the classical harmonic wave theory in mid-latitudes, the meridional structure of the waves' amplitude is described by harmonic functions, i.e., sine, cosine, or exponential functions. This simple theory provides accurate wave solutions when the Coriolis frequency is assumed constant on a plane tangential to the spherical Earth at some latitude  $\phi_0$  (i.e.,  $f = 2\Omega \sin{(\phi)} \approx f_0 = 2\Omega \sin{(\phi_0)}$ , where  $\Omega$  is Earth's frequency of rotation). This model is referred to as the f-plane approximation. In contrast, when the Coriolis frequency is assumed to vary linearly with the meridional coordinate y (i.e.,  $f = f_0 + \beta y$ , where  $\beta = 2\Omega \cos{(\phi_0)}/R$  is constant, where R is Earth's mean radius), the model is referred to as the  $\beta$ -plane approximation. On the  $\beta$ -plane, the harmonic wave theory provides only approximate solutions. A detailed derivation of mid-latitude harmonic waves can be found in the textbooks mentioned earlier in this section. Note that this harmonic structure of waves in mid-latitudes differs substantially from that on the equatorial  $\beta$ -plane where  $\phi_0 = 0$  which yields wave structure that is described by the Hermite functions (Matsuno, 1966) that are not a limiting case of the harmonic structure when  $\phi_0 \to 0$ .

Several observational and numerical studies highlight the limitations of the harmonic wave theory in accurately describing the basic features of mid-latitude Rossby waves. For example, Chelton and Schlax (1996) and Osychny and Cornillon (2004) demonstrate that the phase speed of observed long Rossby waves is greater than that of harmonic Rossby waves, with the difference in phase speeds increasing with latitude. Consistent with the observations, Aoki et al. (2009) used a high-resolution ocean general circulation model (OGCM) and showed that the phase of the simulated Rossby waves propagates faster than predicted by the harmonic wave theory.

An alternate theory, the trapped wave theory, was recently developed for both Poincaré and Rossby waves in wide domains on the mid-latitude  $\beta$ -plane [Paldor et al. (2007); Paldor and Sigalov (2008); Paldor (2015), see details in Appendix A below]. These waves are called trapped since, in contrast to the harmonic waves, they are not spread over the entire meridional domain. Instead, they decay monotonically with latitude from their single maximum located near the equatorward boundary for low modes. The relevance of the trapped wave theory to the ocean was confirmed by satellite observations in the Indian Ocean (De-Leon and Paldor, 2017). Idealized numerical simulations carried out in Gildor et al. (2016) and Yacoby et al. (2023) demonstrate that harmonic wave theory provides accurate approximations for waves only in domains of a small meridional extent, while trapped wave theory does so in large meridional domains. The results reported by Yacoby et al. (2023) also show that the transition from small to large extent depends on the meridional wave mode. Thus, the distinction between "small" and "large" domains is unclear in the context of initial value problems that involve the superposition of several wave modes.

The present study examines the applicability of the harmonic and trapped wave theories to zonally-invariant simulations on the mid-latitude  $\beta$ -plane. Both theories provide valuable but distinct perspectives. The harmonic wave theory, formulated with a constant Coriolis parameter, requires two rigid meridional boundaries to support standing modes and can also be applied locally through WKBJ-type approximations that use a local dispersion relation. These local interpretations are widely used and enhance the applicability of harmonic theory in geophysical contexts. In contrast, the trapped wave theory requires only

a single boundary and yields solutions that decay poleward of this boundary, extending the applicability of linear wave theory to wide meridional domains where harmonic modes do not prevail. Our goal is to systematically compare the accuracy of the two theories by comparing them with numerical simulations and to clarify the parameter regimes where each provides reliable approximations.

The examination is carried out by deriving harmonic and trapped depth-independent wave solutions to two known physical problems and comparing these solutions with the temporal evolution in numerical simulations of a single layer ocean. The physical problems considered here are the geostrophic adjustment problem [see, e.g., Gill (1976, 1982), Blumen (1972) and Yacoby et al. (2021, 2023, 2024)] and the Ekman adjustment problem that results from the addition of a constant zonal wind forcing to the RSWE [see, e.g., Charney (1955), Gill (1982, Sec. 10.9) and Yacoby et al. (2024)]. In both problems, the waves are the key mechanism that transforms the unbalanced initial state to a balanced (i.e., steady) final state. However, the forces that drive the waves are different in the two problems. In the geostrophic adjustment problem, the waves are generated by an initial disturbance (sea surface height anomaly in the case discussed here), while in the Ekman adjustment problem, the waves are generated by wind stress. The assumption of no zonal variations eliminates the Rossby and Kelvin waves from the problem, leaving Poincaré waves (which have not been studied as intensively as Rossby waves) as the sole wave type on which the present study focuses. Under this assumption, the harmonic wave solutions on the  $\beta$ -plane are identical to those on the  $\beta$ -plane. Thus, a comparison between harmonic and trapped wave theories can also be interpreted as a comparison between exact wave solutions on the  $\beta$ -plane are

The paper is organized as follows. Sec. 2 presents the governing equations and the set-up of the geostrophic adjustment and the Ekman adjustment problems. Sec. 3 briefly outlines the harmonic and trapped wave solutions to these problems. Sec. 4 compares these analytic solutions with idealized single-layer ocean simulations and in. Sec. 5 we discuss the results and their implications. The paper also includes five appendices that provide additional technical or side details. Appendix A summarizes the harmonic and trapped solutions of the eigenvalue equation for the meridional velocity from which the particular solutions of Sec. 3 are derived. The corresponding solutions of zonal velocity and sea surface height are provided in Appendix B. Appendix C discusses the relevance of depth-independent (harmonic and trapped) wave solutions to simulations of a two-layer ocean model. In Appendix D, the idealized single-layer ocean simulations of Sec. 4 are compared with nonlinear simulations of a multilayered stratified ocean. Finally, Appendix E addresses the relevance of the idealized harmonic and trapped wave solutions to observations.

#### 2 Set-up of the problems

60

The two physical problems studied in this work – the geostrophic adjustment and the Ekman adjustment – share a common mathematical set-up in the homogeneous part of the differential equations and in the boundary conditions. In contrast, the inhomogeneous term of the differential equation and the initial conditions differ in the two problems. The details of the mathematical set-up in each of these problems are described in this section.

#### 2.1 Governing equations

The zonally invariant, vertically averaged linearized RSWE in a surface layer of mean uniform thickness H forced by a constant (in time and space) zonal wind stress,  $\tau_0$ , are:

$$\frac{\partial u}{\partial t} - f(y)v = \frac{\tau_0}{\rho H},\tag{1}$$

$$\frac{\partial v}{\partial t} + f(y)u = -g\frac{\partial \eta}{\partial y},\tag{2}$$

$$\frac{\partial \eta}{\partial t} + H \frac{\partial v}{\partial y} = 0,\tag{3}$$

where u and v are the vertically averaged velocity components along the x (zonal) and y (meridional) coordinates, respectively,  $\eta$  is the deviation of the fluid height from its mean value H,  $\rho$  is the fluid density, and g is the gravitational acceleration (or the reduced gravitational acceleration when the fluid is stratified). As mentioned above, on the mid-latitude  $\beta$ -plane the Coriolis frequency is given by:

$$f(y) = f_0 + \beta y = 2\Omega \left( \sin(\phi_0) + \frac{\cos(\phi_0)}{R} y \right) \tag{4}$$

where R and  $\Omega$  are Earth's mean radius and frequency, respectively [see, e.g., Gill (1982, Sec. 12.2), Pedlosky (1987, Sec. 3.17 and Chapter 6), Cushman-Roisin and Beckers (2011, Sec 9.4), and Vallis (2017, Sec. 2.3)].

## 2.2 Domain configuration and boundary conditions

The study of wave solutions of the zonally invariant (x-independent) linearized RSWE equations (1)-(3) in a meridional domain,  $y \in [0, L]$ , where L is the domain's meridional extent requires the application of boundary conditions. In both problems, the boundary conditions at the domain's boundaries are the vanishing of the normal velocities, i.e.:

$$v(y=0) = 0 = v(y=L) \tag{5}$$

# 2.3 Initial conditions and wind forcing

105

In both problems, the fluid is assumed to be initially at rest, i.e.:

$$u = 0 = v$$
 at  $t = 0$ . (6)

In the geostrophic adjustment problem, the wind stress,  $\tau_0$  on the RHS of Eq. (1) is set to zero and the initial surface height disturbance is given by:

$$\eta = -\eta_0 \operatorname{sgn}(y - y'), \tag{7}$$

where  $\eta_0$  is the initial disturbance amplitude, sgn(z) is the sign function, and y' is the initial location of the initial discontinuity (front) in fluid height, i.e.:

115 
$$\eta(t=0) = \begin{cases} +\eta_0, & \text{for } 0 \le y 

Figure 1. The meridional velocity, v(y,t) = v', in the geostrophic adjustment problem for L = 4. Black lines: Numerical simulations; Red lines: Analytical harmonic waves; Blue lines: Analytical trapped waves.

**Figure 2.** As in Fig. 1 but for L = 60.

Figure 3. The time-dependent component of the meridional velocity,  $v'=v(y,t)-\bar{v}(y)$ , in the Ekman adjustment problem for L=4. Black lines: Numerical simulations; Red lines: Analytical harmonic waves; Blue lines: Analytical trapped waves.

**Figure 4.** As in Fig. 3 but for L = 60.

also observe a difference in the amplitudes of harmonic and simulated waves at the wave-fronts. The wave-fronts of harmonic waves are larger and sharper compared to those obtained from the simulations, which is particularly noticeable near the domain boundaries at t = 18,30,42, and 54, and at the center of the domain at t = 48. This difference between the theory and the simulations is likely due to the dissipation applied in the MITgcm that reduces the energy contained in the short wave limit. At lower resolutions of both y and t the gap between the theory and simulation at t = 48 occurs earlier and the gap at t = 48 is larger by a factor of about 2.

Fig. 2 shows v' in the geostrophic adjustment problem for L=60. The harmonic wave solutions (red lines) differ substantially from the numerical results (black lines), except near the wave-fronts. In contrast, there is a very good agreement between the trapped wave theory (blue lines) and the numerical results up to t=30. At t=30, the wave-fronts reach the domain boundaries and are reflected towards the center of the domain. This reflection is observed in the numerical results and the harmonic wave solutions. However, in the trapped wave solutions, the waves are reflected only from the southern wall (at y=0). Consequently, a discrepancy between the trapped wave structure and that of the numerical results develops near the northern wall and propagates southward at the speed of the wave-fronts that equals 1 in non-dimensional units (i.e.  $\sqrt{gH}$  in dimensional units, since  $R_d f_0 = \sqrt{gH}$ ). This is evident, for example, at t=48, at which time the northern wave-front, that had reached the northern wall at t=30, is located at y=42=60-18. Thus, the trapped wave theory yields incorrect results between y=42 and y=60. Regardless of the reflection, a small, yet, noticeable difference can be observed between the trapped wave theory and the numerical results, particularly for  $t\geq 24$  and near the center of the domain. We hypothesize that this difference arises from a slight difference between the trapped frequencies and the numerical frequencies.

Fig. 3 shows v' in the Ekman adjustment problem for L=4. As in the geostrophic adjustment problem (Fig. 1), the agreement between the harmonic waves (red lines) and the simulations (black lines) is good, though, as in Fig. 1, a discrepancy is evident between the harmonic and numerical frequencies. In this case, the discrepancy is particularly noticeable at t=36 and t=48. As expected, the trapped wave structure (blue lines) is irrelevant to the simulations at L=4.

Fig. 4 shows v' in the Ekman adjustment problem for L=60. As in the geostrophic adjustment problem (Fig. 2), the harmonic wave solutions (red lines) differ substantially from the numerical results (black lines). For  $t \le 18$ , the discrepancy between the harmonic wave theory and the numerical results is more significant in the northern side of the domain than in its southern side. This may be related to the fact that the term -2by, which is ignored in the harmonic wave theory, increases linearly with y. The trapped wave theory (blue lines) matches the numerical results only for small t. As in the geostrophic adjustment problem, the mismatch between the theory and the simulations develops at the northern wall and spreads southwards. However, in the Ekman adjustment problem, this southward spread begins at t=0. Consequently, in the Ekman adjustment problem, the trapped wave theory provides reasonable results for shorter times compared to the geostrophic adjustment problem. For example, in the Ekman adjustment problem the trapped wave theory yields reasonable results at t=48 only for y

Figure 5. Temporal evolution of the errors. Dotted lines: The time-dependent difference between the wave theories and the numerical solutions,  $\epsilon(t)$  defined in Eq. (27). Red dotted lines: Harmonic wave theory. Blue dotted lines: Trapped wave theory. Solid lines: Low-pass filter of the corresponding dotted lines –  $\epsilon^{LP}(t)$ .

**Figure 6.** Contours of low-pass filtered  $\epsilon$ ,  $\epsilon^{LP}$ , on the t and L plane in the two physical problems and for the two wave theories.

#### **Summary and Discussion** 5







This work examined the applicability of two wave theories on the mid-latitude  $\beta$ -plane – the harmonic and the trapped wave theories – to the temporal evolution evidenced in numerical simulations. The examination is based on the derivation of onedimensional, zonally-invariant, wave solutions for two physical problems – the geostrophic adjustment and the Ekman adjustment problems. The analytical solutions are then compared to numerical simulations conducted using the MITgcm. The numerical simulations are assumed to be accurate and the aim in comparing the theories with numerical simulations is to 365 evaluate the applicability of the idealized theories, rather than the accuracy of the simulations.

The discrepancies between the two theories and numerical simulations were quantified using  $\epsilon(t)$ , defined in Eq. (27), focusing on its low-pass filtered,  $\epsilon^{LP}(t)$ . The discrepancies originate from different approximations associated with each theory. The harmonic wave theory, which neglects the  $\beta$  effect, becomes less accurate when the meridional domain, L, increases to L=20, with more complex variations beyond this domain size (upper panels of Fig. 6). On the other hand, the trapped wave solutions of the geostrophic adjustment problem, that account consistently for  $\beta$ , neglect the family of eigenfunctions associated with the second Airy function – Bi are more accurate as L increases as they better satisfy the boundary condition at the north wall with the increase in L (lower-left panel of Fig. 6). However, in the Ekman adjustment problem, optimal agreement occurs near  $L \approx 30$  (lower-right panel of Fig. 6). Intuitively, the increase of  $e^{LP}$  with L for L > 30 can be attributed to the larger number of wave modes required to accurately describe the solution in large domains, while the number of wave modes used here was identical at all values L. To test this hypothesis,  $\epsilon^{LP}$  in the Ekman adjustment problem was recalculated with the number of summed modes equal to  $10^3$  and  $5 \times 10^4$  (whereas the number of modes used throughout was  $10^4$ ). Contrary to intuition, the effect on  $e^{LP}$  of the change in the number of summed wave modes was insignificant for small t and practically 0 for large t.

Our results clearly demonstrate the failure of the trapped wave theory in small domains. This failure is attributed to two reasons. The main reason, which plays a role in both problems, is that the Airy functions Ai(y) can not satisfy the boundary condition of v'=0 at y=L when L is small. The second reason for the failure is that the superposition of Ai(y) modes fails to satisfy the initial conditions of v. In both cases Bi(y) must be added to the solution in order to satisfy the boundary condition at y = L or the initial condition of v. This reason contributes to the failure of the trapped wave theory only in the geostrophic adjustment problem. The failure of the Ai(y) modes to satisfy the initial condition (19) at small domains is demonstrated in Fig. 7 where  $\frac{\partial v'}{\partial t}(t=0) = \sum_{n=0}^{N} \omega_n a_n^* \hat{v}_n^*(y)$  is shown for the harmonic waves (red lines) and trapped waves (blue lines) for L=4 (left panel) and L=60 (right panel). As in Figs. 1-4, the number of summed-up modes, N, is set to 500 in the expansion to harmonic waves and to  $10^4$  in the expansion to trapped waves. Except for the blue curve on the left panel, all curves accurately approximate  $\partial v'/\partial t(t=0)=2\delta(y-y')$  as is evident from the values of the integrals over the curves that should be 2.0 for a Dirac delta function. The calculated values of the integrals (indicated in the figure using red and blue legends) are close to 2. The largest deviation, of about 5%, occurs for trapped waves in small domains which is evident in the blue curve (and associated legend) on left panel. In contrast to the geostrophic adjustment problem, in the Ekman adjustment problem, the superposition of Ai modes satisfies the initial condition (24) even when L=4, as illustrated in the upper-left panel of Fig. 3.

Figure 7. The derivative of v' with respect to t at t=0 in the geostrophic adjustment problem. Left panel: L=4. Right panel: L=60. Red lines: Harmonic waves. Blue lines: Trapped waves. The ordinate of the left panel is truncated at 60, though the maximal value of the red curve is 184, to ensure the finite values of both curves at  $y \neq 2$  can be clearly seen. According to Eq. (19), the curves should satisfy  $\partial v'(t=0)/\partial t = 2\delta(y-y')$  so the area under the curves should be 2.00. The areas under the red and blue curves are noted in the figure using red and blue legends, respectively.

In large domains, the harmonic theory does not reproduce several features of the simulations, primarily because of the omission of the  $\beta$  effect (recall: Rossby waves are filtered out by the k=0 assumption) rather than from the limited number of summed harmonic modes (500 compared to  $10^4$  Airy modes). This conclusion is evident upon comparisons with the f-plane simulations, where the summation over 500 harmonic modes produces accurate results, confirming that the harmonic wave theory effectively describes the f-plane dynamics (results not shown). Errors in the harmonic solutions also stem from the inclusion of modes with tiny amplitudes in the summation, especially in the Ekman adjustment problem, where the superposition of harmonic modes fails to satisfy the initial condition for v', Eq. (24). These errors are less pronounced in the geostrophic adjustment problem but still affect the wave-front amplitudes. Nevertheless, it should be noted that the harmonic theory can be extended locally through WKBJ-type approximations, which account for the slow meridional variation of the Coriolis parameter by using a local dispersion relation. This local interpretation has been widely applied in geophysical contexts and provides additional validity to the harmonic framework beyond the strict global solutions considered here.




Although both problems share the same governing equation, Eq. (14), their forcing mechanisms are different. In the geostrophic adjustment problem waves are driven by localized initial perturbations and for small t 

Figure 8. The range of L and t for which the harmonic and trapped wave theories yields  $\epsilon^{LP} < 0.1$ . White: Both theories. Red: Only the harmonic wave theory. Blue: Only the trapped wave theory.

In both problems, the discrepancies between the theories and the simulations increase with time. However, for large values of L the error of the harmonic wave theory is larger in the Ekman adjustment problem than in the geostrophic adjustment problem (compare the ordinate ranges of the lower panels of Fig. 5). Part of the reason for the higher values of  $\epsilon^{LP}(t)$  in the Ekman adjustment problems arises from the higher amplitude of the waves themselves in the Ekman adjustment problem compared to the geostrophic adjustment problem (compare the ordinate range of Fig. 2 to that of Fig. 4).

Figure 8 summarizes the ranges of L and t where the theories yield acceptable results, defined by  $\epsilon^{LP} < 0.1$ . The colors in Fig. 8 indicate which theory satisfies  $\epsilon^{LP} < 0.1$  as a function of L and t using the following color codes: White: both theories. Red: harmonic wave theory. Blue: trapped wave theory. Black: neither theory. Regions in which neither theory is accurate are wider in the Ekman adjustment problem, reflecting the greater challenges of modeling its dynamics. In both problems, the trapped wave theory yields  $\epsilon^{LP} < 0.1$  over larger ranges of L and L compared to the harmonic wave theory. As evident from the white regions near the ordinates of Fig. 8, both theories satisfy  $\epsilon^{LP} < 0.1$  for sufficiently small L. This is because the superposition of harmonic and trapped wave modes in the two problems was selected such that the resulting functions satisfy the initial conditions. The failure of the trapped wave theory at large L in the Ekman adjustment problem does not result from the small number of modes, as a change in the number of modes (to  $L^{10}$ ) has a negligible effect on  $L^{10}$ . This delicate issue is left for future study.


This study expands on earlier works by examining the accuracy of wave theories across both time and domain ranges (L-values), rather than focusing solely on two values of L (one small and one large) as was done in Gildor et al. (2016) and Yacoby et al. (2023). It demonstrates that neither of the existing wave theories provides accurate approximations for the waves at all (large) times. This underscores the need for a more comprehensive theory that incorporates the  $\beta$  effect while fully satisfying the boundary conditions. An approach to achieve this goal is to decompose the initial conditions into the basis of the two Airy functions, Ai and Bi, while satisfying the boundary conditions, based on solutions of the transcendental equations that currently have no known explicit solutions.

This paper focuses on zonally-invariant Poincaré waves. However, the approach employed here can also be applied to zonally-dependent problems, e.g., geostrophic adjustment in rotating channels [Gill (1976, Sec. 9), Hermann et al. (1989), Tomasson and Melville (1992), and Yacoby et al. (2023, Sec. 5)], geostrophic adjustment in closed basins (Johnson and Grimshaw, 2014), and wind-driven circulation in closed basins [Pedlosky (1965), Pierini (1998), Sura et al. (2000), LaCasce (2000), Cessi and Primeau (2001), and Cessi and Louazel (2001)]. The extension of this work to a zonally-dependent setup, where Rossby waves are also excited, is left for future works.

Code availability. The MITgcm is described in Marshall et al. (1997) and is available at: https://github.com/MITgcm/MITgcm.git. The input files containing the model configuration and parameters used in this paper are available at: https://doi.org/10.5281/zenodo.14585128 (Yacoby, 2025)

# 440 Appendix A: Harmonic- and trapped-wave theories

This appendix reviews the two types of wave solutions of Eq. (15). We start with classical harmonic waves in Sec. A1 and proceed to trapped waves in Sec. A2. In addition to the solutions for v (that are the main focus of this work) we also provide, for completeness of presentation, the solutions for  $\eta$  and u in Appendix B.

#### A1 Harmonic waves


Although the classical harmonic wave theory is well-known, its discussion here serves to highlight the differences between this wave type and the trapped waves presented in Sec. A2.

In the harmonic theory, the y-dependent term -2by is neglected in Eq. (15). Considering the boundary conditions (5), the resulting equation is solved by the harmonic eigenfunctions:

$$\hat{v}_n = a_n \sin\left[\frac{\pi(n+1)}{L}y\right], \qquad n = 0, 1, \dots$$
(A1)

and the associated eigenvalues:

$$E_n = \left(\frac{\pi(n+1)}{L}\right)^2, \qquad n = 0, 1, \dots$$
 (A2)

The coefficients  $a_n$  are determined in Sec. 3 based on the initial conditions. Substituting the expression for  $E_n$  in Eq. (17) yields the dispersion relation for harmonic Poincaré waves:

$$\omega_n^2 = 1 + \left(\frac{\pi(n+1)}{L}\right)^2. \tag{A3}$$

Before moving on to the trapped wave theory we define the normalized harmonic eigenfunctions:

$$\hat{v}_n^*(y) = \sqrt{\frac{2}{L}} \sin\left[\frac{\pi(n+1)}{L}y\right],\tag{A4}$$

in which the coefficient  $\sqrt{\frac{2}{L}}$  guarantees that:

$$<\hat{v}_n^*, \hat{v}_n^*> = \int_0^L \left(\hat{v}_n^*(y)\right)^2 \mathrm{d}y = 1.$$
 (A5)

The definition of  $\hat{v}_n^*$  is employed in Sec. 3 to determine the coefficient  $a_n$  in Eq. (A1).

Note that in the absence of zonal variations, the harmonic wave solutions are identical to those on the f-plane.

#### A2 Trapped waves

This section presents the trapped wave theory, in which the harmonic wave functions of Section A1 are replaced by Airy functions, as has been shown by Paldor and Sigalov (2008), De-Leon and Paldor (2011), Gildor et al. (2016), and Yacoby et al. (2023).

In the trapped wave theory, Eq. (15) is transformed to an Airy equation:

$$\frac{\mathrm{d}^2 \hat{v}}{\mathrm{d}z^2} - z\hat{v} = 0. \tag{A6}$$

by defining

$$z(y) = -(2b)^{-2/3} [E - 2by].$$

The general solution of (A6) is a linear combination of Ai(z), that decays (faster than exponential) for z > 0, and Bi(z), that 470 grows (faster than exponential) for z > 0, namely:

$$\hat{v} = aAi(z) + bBi(z),\tag{A7}$$

where the coefficients a and b are determined from the initial and/or boundary conditions.

#### A2.1 Semi-infinite domains

In semi-infinite domains  $(L \to \infty)$ , the boundary condition that v vanishes at infinity implies that the coefficient of Bi (that grows to infinity) in Eq. (A7) must be 0. Accordingly, using the definition of z(y), Eq. (A7) reduces to:

$$\hat{v} = a \operatorname{Ai} \left( -(2b)^{-2/3} [E - 2by] \right). \tag{A8}$$

The final step is the application of the wall boundary condition at y=0 i.e. setting z(y=0) in the expression of the  $n^{\text{th}}$  zero of Ai(z), denoted as  $\xi_n$ , e.g.,  $\xi_0=-2.338$ ,  $\xi_1=-4.088$ , etc. (note that  $\xi_n$  are all negative since Ai(z) vanishes only at finite negative values of z). This condition determines the discrete wave functions:

$$\hat{v}_n = a_n \text{Ai} \left[ (2b)^{1/3} y + \xi_n \right]$$
 (A9)

with the corresponding eigenvalues:

$$E_n = -\xi_n (2b)^{2/3}. \tag{A10}$$

Substituting this expression for  $E_n$  in Eq. (16) yields the following dispersion relation for trapped Poincaré waves:

$$\omega_n^2 = 1 - \xi_n (2b)^{2/3}$$
. (A11)

As in Sec. A1 we define the normalized (Airy) eigenfunctions:

$$\hat{v}_n^*(y) = \left[ \frac{2^{\frac{2}{3}}}{2b^{\frac{1}{3}}} \text{Ai}'(\xi_n)^2 \right]^{-1/2} \text{Ai} \left[ (2b)^{1/3} y + \xi_n \right]$$
(A12)

where Ai'(z) is the derivative of Ai(z). The coefficient of Ai(z) in Eq. (A12) guarantee that:

$$<\hat{v}_n^*, \hat{v}_n^*> = \int_{0}^{\infty} (\hat{v}_n^*(y))^2 dy = 1.$$

Note that here the upper bound of the integral is  $\infty$  [and not L as in Eq. (A5)] since the trapped wave modes, Ai(z), vanish at 490 infinity. The form of  $\hat{v}_n^*$  given in Eq. (A12) is employed in Sec. 3 to determine the coefficient  $a_n$  in Eq. (A9).

# A2.2 Large finite domains


Since all Airy wave solutions in Eq. (A9) decay to 0 at large y, these solutions can be expected to apply at sufficiently large, finite, y-domains and not only to semi-infinite domains. Indeed, Paldor and Sigalov (2008); Gildor et al. (2016), and Yacoby et al. (2023) demonstrate that the trapped wave theory provides an accurate approximation for the waves when the domain length, L, is large enough e.g. when:

$$L > (2b)^{-\frac{1}{3}}(2+\xi_n) \tag{A13}$$

which guarantees that z(y=L) > 2 so  $\mathrm{Ai}\big(z(y=L)\big) < 0.035$  which is sufficiently small to justify the neglect of  $\mathrm{Bi}(z)$ . The above constraint on L indicates that the higher the wave mode, n (and with it, the absolute value of  $\xi_n$ ), the larger the domain should be for the trapped wave theory to remain valid. However, this condition completely ignores the time variable, which may also affect the applicability of the trapped wave theory in large but finite domains.

The condition (A13) points to the combined dependence of the  $\beta$ -effect on the domain extent L and  $R_d$ . Though the condition applies to the transition from the harmonic (i.e. the f-plane) wave solutions to the trapped (Airy) wave solutions, it's implication is wider and the effect of  $\beta$  on the f-plane dynamics is determined by both L and  $R_d$  as was shown in Yacoby et al. (2024).

# Appendix B: The solutions of $\eta$ and u

For completeness of presentation, this appendix provides the solutions for  $\eta$  and u. We start with the geostrophic adjustment problem in Sec. B1 and proceed to the Ekman adjustment problem in Sec. B2.

# **B1** Geostrophic adjustment

In the geostrophic adjustment problem,  $\eta$  and u can be divided into time-independent components  $(\bar{\eta}, \bar{u})$  and time-dependent components  $(\eta', u')$ .

## 510 **B1.1** Time-independent components

According to Eq. (10), the time-independent components  $\bar{\eta}$  and  $\bar{u}$  satisfy the geostrophic balance:

$$(1+by)\bar{u} = -\frac{\partial\bar{\eta}}{\partial y}. ag{B1}$$

However, an additional equation must be derived to find  $\bar{\eta}$  and  $\bar{u}$ . The derivation of this additional equation outlined here follows the approach presented in Yacoby et al. (2023). Substituting the continuity equation, Eq. (11), into the y derivative of Eq. (9) yields:

$$\frac{\partial q}{\partial t} = bv, \quad q = \frac{\partial u}{\partial y} + (1 + by)\eta.$$
 (B2)

Substituting the continuity equation once again but this time into the y derivative of Eq. (B2), yields:

$$\frac{\partial}{\partial t} \left( \frac{\partial q}{\partial y} + b \eta \right) = 0. \tag{B3}$$

This conservation equation indicates that the combination of time-dependent variables within the bracket at time t equals their initial combination. The initial conditions (6)-(7) imply:

$$q(t=0)=-(1+by)\operatorname{sgn}(y-y'),$$

and substituting this relation into the time integral of Eq. (B3) yields:

$$\frac{\partial^2 u}{\partial y^2} + (1+by)\frac{\partial \eta}{\partial y} + 2b\eta = -2(1+by)\delta(y-y') - 2b\operatorname{sgn}(y-y'). \tag{B4}$$

The system (B1) and (B4) can be solved numerically by imposing the relevant boundary conditions (see discussion in Yacoby et al. (2023)) and utilizing a standard BVP solver.

#### B1.2 waves


After finding v', the wave components of u and  $\eta$ , u' and  $\eta'$ , can be obtained by substituting v' into Eqs. (9) and (11), respectively, and integrating these equations with respect to time. This results in:

$$u' = -(1+by)\sum_{n=0}^{\infty} \frac{a_n^*}{\omega_n} \hat{v}_n^*(y) \cos(\omega_n t)$$
(B5)

and

$$\eta' = \sum_{n=0}^{\infty} \frac{a_n^*}{\omega_n} \frac{\mathrm{d}\hat{v}_n^*}{\mathrm{d}y} \cos(\omega_n t) \tag{B6}$$

where:

$$\frac{\mathrm{d}\hat{v}_n^*}{\mathrm{d}y} = \frac{\pi(n+1)}{L} \sqrt{\frac{2}{L}} \cos\left[\frac{\pi(n+1)}{L}y\right],$$

according to the harmonic wave theory, and:

$$\frac{\mathrm{d}\hat{v}_{n}^{*}}{\mathrm{d}y} = (2b)^{1/3} \left[ \frac{2^{\frac{2}{3}}}{2b^{\frac{1}{3}}} \mathrm{Ai}'(\xi_{n})^{2} \right]^{-1/2} \mathrm{Ai}' \left[ (2b)^{1/3} y + \xi_{n} \right],$$

according to the trapped wave theory.

# **B2** Ekman adjustment

The calculated solutions of  $\bar{v}$  and v' yields  $\eta$  and u as follows: The substitution of  $v = \bar{v} + v'$  in Eq. (11) and integration with respect to t yields:

$$\eta = \bar{\eta} \cdot t + \eta'$$
 (B7)

where:

$$\bar{\eta} = -\frac{\mathrm{d}\bar{v}}{\mathrm{d}y}, \quad \eta' = -\sum_{n=0}^{\infty} \frac{a_n^*}{\omega_n} \frac{\mathrm{d}\hat{v}_n^*}{\mathrm{d}y} \sin(\omega_n t).$$
 (B8)

Substituting  $v = \bar{v} + v'$  in Eq. (9) yields:

$$u = \bar{u} \cdot t + u' \tag{B9}$$

in which:

$$\bar{u} = \frac{1}{1 + by} \frac{d^2 \bar{v}}{dy^2}, \quad u' = (1 + by) \sum_{n=0}^{\infty} \frac{a_n^*}{\omega_n} \hat{v}_n^*(y) \sin(\omega_n t). \tag{B10}$$

The  $\bar{u} \cdot t$  term solves solves the inhomogeneous part of Eq. (9), i.e.:

$$\bar{u}-(1+by)\bar{v}=1,$$

which is equivalent to Eq. (13). The u' component solves the homogeneous part of Eq. (9), i.e.:

$$\frac{\partial u'}{\partial t} - (1 + by)v' = 0.$$

# Appendix C: Extension to a two-layer ocean

In this appendix, we consider the case of a two-layer ocean. This analytically tractable configuration provides the motivation for the continuously stratified case discussed in Appendix D. To this end, the zonally invariant, linearized RSWE (1)–(3) are extended to the two-layer system. For the top layer of mean depth  $H_1$  (with variables denoted by the subscript 1), the governing equations are:

$$\frac{\partial u_1}{\partial t} - f(y)v_1 = \frac{\tau_0}{\rho_1 H_1},\tag{C1}$$

$$\frac{\partial v_1}{\partial t} + f(y)u_1 = -g\frac{\partial \eta}{\partial y},\tag{C2}$$

$$\frac{\partial h}{\partial t} - H_1 \frac{\partial v_1}{\partial y} = 0,\tag{C3}$$

where f(y) is given in (4),  $\eta$  is the free surface displacement and h is the (upward) displacement of the interface that separates the two layers. The continuity equation (C3) assumes  $\left|\frac{\partial \eta}{\partial t}\right| \ll \left|\frac{\partial h}{\partial t}\right|$ , an assumption referred to as the rigid lid approximation. For the lower layer (variables denoted by the subscript 2), the equations are:

$$\frac{\partial u_2}{\partial t} - f(y)v_2 = 0, (C4)$$

$$\frac{\partial v_2}{\partial t} + f(y)u_2 = -g\frac{\partial \eta}{\partial y} - g'\frac{\partial h}{\partial y},\tag{C5}$$

$$\frac{\partial h}{\partial t} + H_2 \frac{\partial v_2}{\partial y} = 0,\tag{C6}$$

where  $H_2$  is the mean thickness of the lower layer and  $g' = g(\rho_2 - \rho_1)/\rho_2$  (where  $\rho_1$  and  $\rho_2$  are the densities of the upper and lower layers, respectively) is the reduced gravity. The momentum equations (C4)-(C5) assume  $(\rho_2 - \rho_1)/\rho_2 \ll 1$  while  $g\frac{\rho_2 - \rho_1}{\rho_2}$  is O(1), an assumption referred to as the Boussinesq approximation. A more detailed derivation of Eqs. (C1)-(C6) can be found in Sec. (9.10) of Gill (1982).

The momentum equations can be combined to eliminate  $\eta$  from the equations, which is achieved by subtracting Eqs. (C1)-570 (C2) from Eqs. (C4)-(C5), respectively. The resulting equations are:

$$\frac{\partial U_{2-1}}{\partial t} - f(y)V_{2-1} = -\frac{\tau_0}{\rho_1 H_1},\tag{C7}$$

$$\frac{\partial V_{2-1}}{\partial t} + f(y)U_{2-1} = -g'\frac{\partial h}{\partial y},,\tag{C8}$$

where:

$$U_{2-1} = u_2 - u_1, \quad V_{2-1} = v_2 - v_1.$$

A continuity equation that involves  $V_{2-1}$  (instead of  $v_1$  or  $v_2$ ) is obtained by adding  $H_1^{-1}$  times (C3) and  $H_2^{-1}$  times (C6), which yields:

$$\left(\frac{1}{H_1} + \frac{1}{H_2}\right) \frac{\partial h}{\partial t} + \frac{\partial V_{2\text{-}1}}{\partial y} = 0,$$

or:

$$\frac{\partial h}{\partial t} + H_e \frac{\partial V_{2-1}}{\partial y} = 0, \tag{C9}$$

where

$$H_e = \frac{H_1 H_2}{H_1 + H_2}.$$

The two-layer system (C7)-(C9) is similar to the single layer system (1)-(3) with two notable differences: (i) The RHS of Eq. (C7) contains a negative sign, whereas the RHS of Eq. (1) does not. (ii) The two-layer system includes two mean heights,  $H_1$  and  $H_2$  (or  $H_1$  and  $H_e$ ), whereas the one-layer system includes only one (H). In other words, the two-layer system introduces an additional free parameter.

#### C1 Nondimensionalization

As in Sec. 2.4, the two-layer system, Eqs. (C7)-(C9), is nondimensionalized (nondimensional variables are denoted by asterisks) by scaling the dimensional variables on:

$$t^* = f_0 t,$$

$$(x^*, y^*) = \frac{1}{R'_d}(x, y), \quad R'_d = \sqrt{g' H_e} / f_0.$$

For the geostrophic adjustment problem (where  $\tau_0 = 0$ ), we also define:

$$h^* = \frac{1}{h_0}h,$$

$$(U_{2-1}^*, V_{2-1}^*) = \frac{H_e}{h_0} \frac{1}{\sqrt{g'H_e}} (U_{2-1}, V_{2-1}),$$

where  $h_0$  the amplitude of the initial interface disturbance (defined in Sec. D2), while, for the Ekman adjustment problem (where  $h_0 = 0$ ) we define:

$$\begin{split} h^* &= \frac{H_1}{H_e} \frac{\rho_1 f_0 \sqrt{g' H_1}}{\tau_0} h, \\ (U_{2\text{-}1}^*, V_{2\text{-}1}^*) &= \frac{\rho_1 f_0 H_1}{\tau_0} \left( U_{2\text{-}1}, V_{2\text{-}1} \right). \end{split}$$

With these nondimensional variables, Eqs. (C7)-(C9) become:

$$\frac{\partial U_{2\text{-}1}^*}{\partial t^*} - (1 + b'y^*) V_{2\text{-}1}^* = -\delta_{i0},\tag{C10}$$

$$\frac{\partial V_{2\text{-}1}^*}{\partial t^*} + (1 + b'y^*) U_{2\text{-}1}^* = -\frac{\partial h^*}{\partial y^*},$$
 (C11)

$$\frac{\partial h^*}{\partial t^*} + \frac{\partial V_{2-1}^*}{\partial y^*} = 0,\tag{C12}$$

where:





$$b' = \frac{\beta R_d'}{f_0}.$$

Note that although the dimensional two-layer system contains more parameters than the dimensional one-layer system, our somewhat cumbersome scaling (compared to that employed in Sec. 2.4) guarantees that the non-dimensional two-layer system contains only one free parameter, exactly as the non-dimensional single layer system.

# Appendix D: Application to simulations with a 3D-OGCM

In this appendix, we extend the analytical insights from Appendix C to realistic simulations using the MITgcm, now employed as a fully 3-dimensional Ocean General Circulation Model, thereby demonstrating the relevance of our results to the real ocean. Although the MITgcm is not inherently a layered model, we configure it with 38 vertical layers to represent a simplified, two-layer physical ocean. The upper and lower physical layers correspond to groups of numerical layers: the lower physical layer is initialized at temperature  $T_1$ , and the upper layer at  $T_2$  (see Sec. D2). Section D2 provides a detailed description of how the numerical layers map onto the physical layers, clarifying that the "physical layers" serve as a conceptual framework for comparison with the two-layer analytical model of Appendix C, while the numerical layers determine the vertical resolution of the 3D-OGCM. Unlike the analytical model, which assumes a sharp interface preventing mixing, the 3D-OGCM includes temperature diffusion, allowing some mixing near the interface (i.e., within the thermocline).

# D1 Equations solved

The MITgcm is employed here to simulate depth-dependent flow with density determined only by temperature. Viscous and diffusive terms are incorporated into the momentum equations and the temperature advection equation, respectively. Similar to the set-up in Sec. 2, the domain is periodic in the zonal direction and bounded in the meridional direction by walls located at y=0 and y=L and aligned parallel to the x-axis. A wind-stress momentum forcing is applied in the zonal momentum equation. However, in this multilayer configuration, the forcing term  $F_{\rm wind}$  is applied only to the momentum equation for the surface layer, i.e., it is set to zero for the interior layers. While the MITgcm model equations account for x-variations, the initial conditions employed here (see Sec. D2) and the periodic boundary conditions in the x-direction ensure that no x-variation develops in the simulations (which was verified by our numerical simulations). Thus, although the equations of the MITgcm include the changes with x, the relevant equations in our problems assume  $\partial/\partial x=0$ . These considerations lead to the following set of equations, written in Cartesian coordinates:

1. Momentum equations:

$$\frac{Du(t,y,z)}{Dt} - f(y)v - A_y \frac{\partial^2 u}{\partial y^2} - A_z \frac{\partial^2 u}{\partial z^2} = F_{\text{wind}}, \tag{D1}$$

$$\frac{Dv(t,y,z)}{Dt} + f(y)u - A_y \frac{\partial^2 v}{\partial y^2} - A_z \frac{\partial^2 v}{\partial z^2} = -\frac{1}{\rho_0} \frac{\partial p'}{\partial y}, \tag{D2}$$

where


$$\frac{D}{Dt} = \frac{\partial}{\partial t} + v \frac{\partial}{\partial y}$$

and  $F_{\text{wind}} = \frac{\tau_0}{\rho_0 \Delta z_s}$  is applied only to the momentum equation for the topmost layer. Here,  $A_y$  and  $A_z$  are horizontal and vertical viscosities, respectively, p is the pressure and  $\rho_0$  is the mean water density [or the reference density in the equation of state, (D5)] and  $\Delta z_s$  is the thickness of the model's topmost layer.

2. Conservation of mass:

$$\frac{\partial \eta(t,y)}{\partial t} + \frac{\partial V}{\partial y} = 0,\tag{D3}$$

where  $\eta$  is the deviation of the sea surface height from z = 0 and  $V = \int v dz$  (i.e. V is the vertically integrated meridional velocity in units of  $m^2/s$ ).

3. Equation for the perturbation pressure, p':

$$p'(t,y,z) = g\rho_0 \eta + \int_z^0 g\rho' dz$$
 (D4)

separated into a barotropic part (due to variations in  $\eta$ ) and a baroclinic part (due to variations in density anomaly,  $\rho'$ ).

4. Linear equation of state:

$$\rho'(t, y, z) = \rho - \rho_0 = -\rho_0 \alpha (T - T_0) \tag{D5}$$

where  $\alpha$  is the thermal expansion coefficient and  $T_0$  is a reference temperature that determines  $\rho_0$ .

5. An advection-diffusion equation for the temperature, T:

$$\frac{DT(t,y,z)}{Dt} - \kappa_y \frac{\partial^2 T}{\partial y^2} - \kappa_z \frac{\partial^2 T}{\partial z^2} = 0$$
(D6)

where  $\kappa_y$  and  $\kappa_z$  are horizontal and vertical diffusivities, respectively. The initial conditions and model parameters are described in Sec. D2.

#### 650 D2 Initial conditions and model parameters

As in the one-layer case discussed in the main text (Sec. 2.3), we consider two types of initial conditions: one for the geostrophic adjustment problem and another for the Ekman adjustment problem. In both problems the ocean is initially at rest, its surface height,  $\eta$ , is zero and it consists of two layers of different temperatures (hence, different densities). The upper (lower) layer has a temperature of  $T_1$  ( $T_2$ ) with  $T_2 < T_1$  and a mean height of  $H_1$  ( $H_2$ ).

In the geostrophic adjustment problem, the forcing term on the RHS of Eq. (D1) is set to zero and the initial interface between the upper and lower layer, h(y,0) is given by:

$$h(y, t = 0) = -h_0 \operatorname{sgn}(y - y') - H_1$$

where  $h_0$  is the amplitude of the initial interface displacement. Accordingly, as illustrated in Fig. D1, the initial temperature field is:

$$T(y < y', z, t = 0) = \begin{cases} T_1, & \text{for } -H_1 + h_0 < z \le 0, \\ T_2, & \text{otherwise,} \end{cases}$$
 (D7)

$$T(y > y', z, t = 0) = \begin{cases} T_1, & \text{for } -H_1 - h_0 < z \le 0, \\ T_2, & \text{otherwise.} \end{cases}$$
 (D8)

while the corresponding initial density anomaly ( $\rho'$ ) field, determined only by the temperature according to the linear equation of state Eq. (D5), is

$$\rho'(y < y', z, t = 0) = \begin{cases} -\rho_0 \alpha(T_1 - T_0), & \text{for } -H_1 + h_0 < z \le 0, \\ -\rho_0 \alpha(T_2 - T_0), & \text{otherwise,} \end{cases}$$
 (D9)

$$\rho'(y > y', z, t = 0) = \begin{cases} -\rho_0 \alpha(T_1 - T_0), & \text{for } -H_1 - h_0 

Figure D1. A schematic illustration of the initial temperature profile in the geostrophic adjustment problem, Eqs. (D7)-(D8)

for y < y' and y > y' to represent an initial disturbance in the thermocline depth. Specifically, for y > y', the upper (physical) layer consists of 23 numerical layers, whereas for y 

Figure D2. The meridional velocity for the geostrophic adjustment problem in multilayered ocean simulations. Red: the vertically averaged velocity of the lower layer,  $v_2$ . Blue: the vertically averaged velocity of the upper layer,  $v_1$ . Dashed-black:  $V_{2-1} = v_2 - v_1$ . Solid-black: the meridional velocity in one-layer ocean simulations,  $v_1$  (duplicates of the solid-black lines shown in Fig. 2).

Figure D3. The time-dependent component of the meridional velocity for the Ekman adjustment problem in multilayered ocean simulations. Red: the vertically averaged velocity of the lower layer,  $v'_2$ . Blue: the vertically averaged velocity of the upper layer,  $v'_1$ . Dashed-black:  $V'_{2-1} = v'_2 - v'_1$ . Solid-black: the time-dependent component of the meridional velocity in one-layer ocean simulations, v' (duplicates of the solid-black lines shown in Fig. 4).

The figures show excellent agreement between  $V'_{2\text{-}1}$  in the multilayered ocean simulations (dashed black) and v' in the simple single-layer ocean simulations (solid black) in both problems. However, in the geostrophic adjustment problem (Fig. D2), discrepancies between  $V'_{2\text{-}1}$  and v' are observed near the wave-fronts, where waves with a relatively short wavelength exist. We hypothesize two reasons for the discrepancies between  $V'_{2\text{-}1}$  and v': (i) The horizontal viscosity terms added to the momentum equations in the 3D-OGCM, Eqs. (D1)-(D2), reduce the energy of short waves in the multilayered ocean, resulting in smoother wave-fronts. (ii) To accelerate the multilayered ocean simulations, we significantly increased  $\Delta t$  and  $\Delta y$  in the 3D-OGCM (compare Table D2 with Table 1). As mentioned in Sec. 4.1, the sharpness of the wave-fronts decreases as  $\Delta t$  and  $\Delta y$  increase. In addition to the agreement between the single layer and multilayered simulations, the figures clearly indicate that in both problems  $v'_1 = -v'_2$  so  $V'_{2\text{-}1} = 2v'_2 = -2v'_1$ .

We conclude this section with results not shown in the figures: (i) In both problems, the velocity in the lower layer is uniform with depth. Thus, the velocity at any depth below the interface equals the vertically averaged velocity of the lower layer,  $v_2'$ . (ii) In the geostrophic adjustment problem, the velocity in the upper layer is uniform with depth, as is the velocity in the lower layer. (iii) In the Ekman adjustment problem, the wind stress (which acts only at the topmost layer) causes a shear of the flow of the upper layer. We found that the profile of v'(z) in the upper layer depends on the thickness of the model's topmost layer,  $\Delta z_s$ . However, the vertically averaged velocity  $v_1'$  is independent of  $\Delta z_s$  since in a thinner grid layer the effect of the wind stress in that layer increases (the same wind stress is spread over a thinner layer).

#### **Appendix E: Relevance to observations**



Due to their relatively fast phase speed, Poincaré waves in the ocean are harder to observe compared to Rossby waves. Yet, reports of Poincaré wave observations were documented in the literature and they have been compared with analytical solutions and numerical simulations. For example, internal Poincaré waves were observed in Lake Ontario following a storm on 9 August 1972. Simons (1978) analyzed these observations and showed that both analytical and numerical solutions in idealized setting exhibit similar characteristics to the observed wave-fronts, e.g., the offshore propagation speed and the periodic recurrence with near-inertial periods. Simons (1978) also showed that the basic kinematics of the downwelling front could be simulated using a simple two-layer model.

Observations of the fast Poincaré waves require long and high-resolution in time and, similarly, the distinction between the mode structure of trapped and harmonic waves requires high meridional resolution and large meridional extent, both of which complicate the detection of these waves in the ocean. Presently, observations of Poincaré waves were reported mainly in lakes, where only harmonic modes can be detected, e.g., Lake Michigan and Lake Ontario (see, e.g., Mortimer, 1977). Indeed, Gill (1982, Sec. 7.3) cites these observations, emphasizing that the observed Poincaré waves have similar characteristics to the analytical harmonic-wave solutions of the geostrophic adjustment on the f-plane. Our results imply that the resemblance between both numerical and analytical solutions on the f-plane and the observed waves in Lake Ontario is expected, given that the meridional (south-north) extent of Lake Ontario is  $\sim 80 \ km$ , which should be considered narrow since the results of Figs. 1 and 3 imply that a meridional extent of O(4) radii of deformation is narrow.

Poincaré waves with frequency near the inertial frequency f, known as near-inertial waves, are a dominant mode of high-frequency variability in the ocean, appearing as a prominent peak that rises significantly above the Garrett and Munk (1975) continuum internal wave spectrum (see, e.g., Alford et al., 2016). These waves are frequently observed in oceans and lakes, such as in the Gulf of Mexico (Gough et al., 2016), Lake Ontario (Schwab, 1977), Lake Michigan (Ahmed et al., 2014), the Gulf of Lions (Millot and Crépon, 1981), and the northeast Pacific Ocean (D'Asaro et al., 1995). The distinction between near-inertial trapped and harmonic modes of these near-inertial waves is complicated by the fact that the frequencies of the n=0 modes are very close to 1, hence to one another. This can be shown by substituting n=0 in Eq. (A3) which yields  $\omega^2=1+(\pi/L)^2$  for the harmonic n=0 mode while substituting n=0 (i.e.,  $\xi_0=-2.338$ ) and b=0.005 in Eq. (A11) yields  $\omega^2=1.1$  for the trapped wave theory. For L=10 the two types of n=0 modes yield identical frequencies and for a larger/smaller value of L the frequency of the harmonic mode is only slightly smaller/larger than that of the trapped mode.

However, the trapped wave solution can be invoked to reproduce an observed phenomenon in the ocean – the linear change of the meridional wavenumber with time. The observations in the Pacific Ocean reported in D'Asaro et al. (1995) demonstrate that following a storm the zonal wavenumber remains constant while the meridional wavenumber changes linearly with time. Specifically, the meridional wavenumber decreases at a rate of  $-\beta t$  by, first, decreasing the initial wavenumber to zero followed by a  $180^{\circ}$  phase shift in which the wavenumber becomes negative and increases its absolute value linearly with time (see also Alford et al., 2016). This phenomenon was explained by D'Asaro et al. (1995) using the following argument: Representing an inertial wave on the  $\beta$ -plane (where  $f = f_0 + \beta y$ ) as  $e^{i(l_0y - ft)} = e^{i[(l_0 - \beta t)y - f_0t]}$  suggests that the initial meridional wavenumber  $l_0$  becomes increasingly negative as  $\beta t$  increases. However, this heuristic argument is mathematically inconsistent since the ansatz  $e^{i(l_0y - f(y)t)}$  violates the separation of variable that yields the wave equation for the meridional structure (and the dispersion relation). Indeed, the harmonic wave solutions (red lines in Figs. 1-4) do not reproduce the linear time variation of the meridional wavenumber.




In contrast to the harmonic wave solutions, the trapped wave theory accurately reproduces the linear change of the wavenumber. To illustrate this, Fig. E1(a) revisits the trapped wave solutions for the Ekman adjustment problem shown in Fig. 4. The two southern nodal points are highlighted with red dots and the distance between these points, D, provides an estimate of the meridional wavenumber –  $l \approx \pi/D$ . As shown in Fig. E1(b), the calculated wavenumber increases linearly with time. A linear regression analysis yields a slope of 0.0051, which is in excellent agreement with the theoretical value  $\beta R_d/f_0 = 0.005$  and the trend observed by D'Asaro et al. (1995).

The application of the theoretical results reported here to observations does not include the meridional structure of the modes and the wave's spectrum, since under typical conditions these properties cannot be deciphered in observations. However, laboratory experiments on a rotating table, similar to those reported in Cohen et al. (2010) and Cohen et al. (2012), can be carried out to verify the applicability of the theoretical results to carefully designed laboratory experiments.

Author contributions. IY: Formal analysis, Investigation, Visualization, Writing the original draft, Reviewing, Editing. HG: Validation, Reviewing, Editing.

Figure E1. The decrease of the meridional wavelength with time. (a) The blue curves replicate the blue curves of Fig. 4, i.e., the trapped wave solutions in the Ekman adjustment problem at the indicated times, t. The red dots mark the two southern nodal points. The distance between the two red dots, D, is used in panel (b) to estimate the meridional wave number,  $l \approx \pi/D$ . (b) Dots: The estimated zonal wavenumber  $\pi/D$  as a function of time. Dashed line: A linear regression fit. The slope of the regression line is  $0.0051 \approx \beta R_d/f_0 = 0.005$  which agrees very well with the observed trend reported by D'Asaro et al. (1995). The intersection with the ordinate is -0.02, indicating that the initial wavelength is 314. A  $180^{\circ}$  phase shift occurs at t=4.

| NP: Conceptualization, I | Investigation. | Methodology, P | roject administration. | Writing the | original draft. | Reviewing, Editing. |
|--------------------------|----------------|----------------|------------------------|-------------|-----------------|---------------------|
|                          |                |                |                        |             |                 |                     |

Competing interests. The authors declare that they have no conflict of interest

*Acknowledgements.* This study was supported by the Joint National Natural Science Foundation of China–Israel Science Foundation (research grant number 2547/17) and by the U.S.–Israel Binational Science Foundation (BSF grant number 2018152).

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
