# Peer review of "On the applicability of linear wave theories to simulations on the mid-latitude $\beta$ -plane"

_EGUsphere, 2025_

## Referee Comment (RC1)

**Referee's Report**

**egusphere-2025-2529**

**Title: On the applicability of linear wave theories to simulations on the mid-latitude $\beta$-plane**
**Authors: Itamar Yacoby, Hezi Gildor, and Nathan Paldor**

**Recommendation:** I recommend minor revisions.

In this paper, two ways to treat fluid motions on mid-latitude $\beta$-plane are described and compared. One is a conventional way to treat mid-latitude $\beta$-plane, which is, when considering inertial gravity waves, essentially the same as f-plane where the Coriolis parameter is represented as a certain constant value $f_0$, and in this paper, it is called "harmonic wave theory". The other is to treat the Coriolis parameter as linearly changing with y or $f = f_0 + \beta y$ also for gravity waves, and it is called "trapped wave theory".

Some numerical calculations or time integrations of a simple $\beta$-plane numerical model are performed, and the accuracies of harmonic wave theory and trapped wave theory are investigated. And the results are summarized such as the error of the harmonic wave theory becomes large when the latitudinal width is large, while the error of the trapped wave theory becoms large when the latitudinal width is small.

The comparison of the two theories for simple $\beta$-plane calculation is important and the results obtained here are also significant; it is worth publishing, I think. However, when I first read this article, I over-hoped for this trapped wave theory; it must overcome the defects of the harmonic wave theory since the change of Coriolis parameter is exactly treated without approximation in the trapped wave theory. But, actually, the situation where this trapped wave theory can be applied is quite limited; the region should be widely spread in the north direction, the region should be limited by some cost in the south direction, the motion should be relatively large so that the frequecy of the wave is near to the intertial cycle, which assure that the motion is evanescent in the north area but wavy in the south, and so on. On the other hand, in the conventional harmoni theory, although the Coriolis parameter is treated as constant, we are usually able to consider "local dispersion relation" which is often used as WKBJ treatment. In this article some real examples of observations or simulations are described, and most of them are referred to as examples which show the advantage of the trapped wave theory. In some examples, the difference of the phase speed or frequency cannot be explained by harmonic theory where the Coriolis parameter is constant, but if we consider local value of the Coriolis

parameter, it may be explained. As another example shown in p.32, the D'Asaro et al's argument is critisized, the presumed form of the wave does not satisfy the wave equation. This becomes a problem when considering exact solution in the entire region, but if we consider approximation as WKBJ treatment and consider local relation, which is possible when considering solutions like harmonic theory, it does not become a problem.

It is true that the harmonic theory has advantage as an exact solution, but harmonic theory has advantage that it can be considered locally like WKBJ treatment. I think that the theoretical and calculation parts in this artilce does not have problems but introduction and discussion parts where the comparison with real observation is desceibed should be reconsidered, since the harmonic theory and earlier theories are unfarily critisized.

---

## Author Comment (AC1)

**Response to Referee #1's comments on egusphere-2025-2529**

by: I. Yacoby, H. Gildor, and N. Paldor

The comments of Referee #1 are quoted below in blue, and the authors' responses are written in black. All references (figures, sections, equation numbers, and line numbers) in the responses refer to the revised manuscript.

**Response to Referee 1**

In this paper, two ways to treat fluid motions on mid-latitude $\beta$-plane are described and compared. One is a conventional way to treat mid-latitude $\beta$-plane, which is, when considering inertial gravity waves, essentially the same as f-plane where the Coriolis parameter is represented as a certain constant value $f_0$, and in this paper, it is called "harmonic wave theory". The other is to treat the Coriolis parameter as linearly changing with y or $f = f_0 + \beta y$ also for gravity waves, and it is called "trapped wave theory".

Some numerical calculations or time integrations of a simple $\beta$-plane numerical model are performed, and the accuracies of harmonic wave theory and trapped wave theory are investigated. And the results are summarized such as the error of the harmonic wave theory becomes large when the latitudinal width is large, while the error of the trapped wave theory becoms large when the latitudinal width is small.

We thank the reviewer for the constructive comments. Below we provide a point-by-point response to the particular points of concern.

The comparison of the two theories for simple $\beta$-plane calculation is important and the results obtained here are also significant; it is worth publishing, I think. However, when I first read this article, I over-hoped for this trapped wave theory; it must overcome the defects of the harmonic wave theory since the change of Coriolis parameter is exactly treated without approximation in the trapped wave theory. But, actually, the situation where this trapped wave theory can be applied is quite limited; the region should be widely spread in the north direction, the region should be limited by some cost in the south direction, the motion should be relatively large so that the frequecy of the wave is near to the intertial cycle, which assure that the motion is evanescent in the north area but wavy in the south, and so on.

We thank the reviewer for this important comment. We agree that the trapped wave theory is not universally applicable, but we note that it is not more restrictive than the harmonic theory. In fact, the two theories differ fundamentally in their boundary requirements. While the harmonic wave theory requires two rigid meridional boundaries to support standing wave solutions, the trapped wave theory requires only a single meridional boundary, since the solutions decay poleward from this single rigid boundary. This distinction implies that the trapped wave theory applies in configurations where the harmonic solutions cannot exist at all. Thus, the trapped wave theory is not more limited, but rather complementary, as it applies to wide domains with only one rigid boundary (located on the south side of the domain in the northern hemisphere).

On the other hand, in the conventional harmonic theory, although the Coriolis parameter is treated as constant, we are usually able to consider "local dispersion relation" which is often used as WKBJ treatment.

We agree that the harmonic theory, although formulated with a constant Coriolis parameter, can often be applied locally using WKBJ-type approximations that rely on a local dispersion relation. This important aspect of the theory broadens its applicability in geophysical contexts and complements the global standing-wave solutions. Following this comment, we revised the discussion section. In the revised version, at the end of the paragraph discussing the failure of the harmonic wave theory in large domains, we added the following clarification:

"Nevertheless, it should be noted that the harmonic theory can also be extended locally through WKBJ-type approximations, which account for the slow meridional variation of the Coriolis parameter by introducing a local dispersion relation. This local interpretation has been widely applied in geophysical contexts and extends the usefulness of the harmonic framework beyond the global solutions considered here."

In this article some real examples of observations or simulations are described, and most of them are referred to as examples which show the advantage of the trapped wave theory. In some examples, the difference of the phase speed or frequency cannot be explained by harmonic theory where the Coriolis parameter is constant, but if we consider local value of the Coriolis parameter, it may be explained. As another example shown in p.32, the D'Asaro et al's argument is critisized, the presumed form of the wave does not satisfy the wave equation. This becomes a problem when considering exact solution in the entire region, but if we consider approximation as WKBJ treatment and consider local relation, which is possible when considering solutions like harmonic theory, it does not become a problem.

It is true that the harmonic theory has advantage as an exact solution, but harmonic theory has advantage that it can be considered locally like WKBJ treatment. I think that the theoretical and calculation parts in this artilce does not have problems but introduction and discussion parts where the comparison with real observation is desceibed should be reconsidered, since the harmonic theory and earlier theories are unfairly critisized.

We agree that in the original submission, the Introduction and Discussion may have conveyed an overly negative impression of the harmonic wave theory when contrasted with the trapped wave theory. Our intention was not to dismiss the usefulness of harmonic theory, but rather to highlight the limitations of its global form relative to trapped modes. Following the reviewer's constructive suggestion, we revised both sections to present a more balanced view and to explicitly acknowledge the contexts in which harmonic theory remains valid. In particular, we now emphasize that the harmonic framework has two complementary aspects: (i) its role as an exact solution on the $f$–plane, and (ii) its ability to serve as a local approximation on the $\beta$–plane through WKBJ-type methods. These additions clarify that many observational results (including those discussed in relation to D'Asaro et al.) can be reconciled with local interpretations of harmonic theory.

**Changes made in the manuscript:**

- **Introduction (Sec. 1):** We added a sentence explicitly noting that the harmonic theory, although requiring two rigid boundaries, can also be applied locally through WKBJ-type approximations, which use a local dispersion relation. This addition clarifies that harmonic theory provides valuable insights not only as a global exact solution but also as a practical local approximation in geophysical contexts.

- **Discussion (Sec. 5):**

  1. We softened the wording where the harmonic theory was described as "failing" in wide domains. The revised text now states that "in large domains, the harmonic theory does not reproduce several features of the simulations."

  2. At the end of the paragraph on the harmonic theory in large domains, we added the following clarification:

     "Nevertheless, it should be noted that the harmonic theory can be extended locally through WKBJ-type approximations, which account for the slow meridional variation of the Coriolis parameter by using a local dispersion relation. This local interpretation has been widely applied in geophysical contexts and provides additional validity to the harmonic framework beyond the strict global solutions considered here."

---

## Author Comment (AC2)

**Response to Referee #2's comments on Manuscript ID egusphere-2025-2529**

by: I. Yacoby, H. Gildor, and N. Paldor

The comments of Referee #2 are quoted below in blue, and the authors' responses are written in black. All references (figures, sections, equation numbers, and line numbers) in the responses refer to the revised manuscript.

**Response to Referee 2**

This manuscript explores two one-dimensonal wave theories for inertia-gravity waves on a mid-latitude beta-plane and consider both geostrophic and Ekman adjustments. They focus mainly on linear theories but also present a few nonlinear simulations. They define a metric to quantify the differences between linear and nonlinear theories and show how this varies in different regimes.

This research is of interesting to the community and well written but I strongly recommend that the following concerns are addressed before it is published.

We thank the reviewer for the constructive comments and provide below a point-by-point response to the particular points of concern.

Please give citations for the particular scalings you are using in the two cases.
Done (see L120 and L125).

The authors work very hard to have one set of equations that allows for the two types of adjustments. This makes things mathematically complicated and I'm not sure how much is gained. If there is a big benefit to this, please emphasize this as it's not entirely clear to me.
We thank the referee for pointing out this issue. We agree that the unified formulation of the geostrophic and Ekman adjustment problems introduces additional algebraic complexity. However, the benefit is that both problems are now cast into a single nondimensional system, and the only difference between the problems appears only in the Kronecker delta $\delta_{i0}$ function on the RHS of Eq. (9). This has two advantages: (i) it allows a direct comparison of the two classical problems in the same mathematical framework, which clearly distinguishes between aspects of the dynamics are problem-specific and those common to both problems; and (ii) it reduces redundancy by avoiding two parallel derivations. To make this benefit clear, we have revised the manuscript and added an explanatory remark at the end of Sec. 2.4.

In section 2.5, going from equation 14 to 15, the quadratic term is ignored. This is because the authors want to get the equation that has a known special function a solution. However, this is an approximation, and introduces more error into the equation. Since the eigenvalue problem is solved numerically, you can easily keep in this term and that should yield a more accurate theory. Please work on doing this or give a very good reason why not.
We thank the referee for this suggestion. The neglect of the quadratic term $b^2y^2$ in Eq. (14) is justified for the following reasons:

1. As pointed out by Paldor and Sigalov (2008), neglecting $b^2y^2$ is consistent with the standard $\beta$-plane approximation, which retains only terms linear in $y$ in the expansion of the Coriolis frequency (Eq. 4) and ignores the quadratic (and higher) terms. To maintain the $O(y^2)$ terms in $f(y)^2$ (e.g. in Eq. (14) where $(1 + by)^2$ is the scaled form of $f(y)^2$) requires that the same terms are also maintained in the expansion of $f(y)$.

2. The errors introduced by this approximation are indeed negligible. This is evident in Fig. 2, where the trapped wave solutions (blue curves) and the numerical solutions (black curves) are almost identical for $t < 30$. After $t = 30$, discrepancies arise from reflections at the domain boundaries, not from the neglect of the quadratic term.

3. The main aim of the paper is to evaluate the accuracy of the two existing wave theories and not to develop a new theory. Retaining the quadratic term would complicate the analytical treatment without improving the conceptual comparison of the known solutions (which were derived based on the procedure described in the present MS).

Thus, the neglect of the $b^2y^2$ term is justified and appropriate for the goals of the present study.

Section 3 discusses harmonic, trapped and semi-infinite domains. These have all been previously studied in the context of the shallow water model. In reality, waves will not be purely harmonic or trapped, but some hybrid of the two. If that's the case, then why spend so much time focusing on each of these limits? Again, with numerical solutions you can study any of these waves and you need not restrict yourselves to these relatively simple cases.

Our focus on harmonic, trapped, and semi-infinite domains is not meant to imply that real waves fall only into these categories. Rather, these limiting cases provide a clear framework for understanding fundamental dynamics, isolating the effects of individual physical processes and for benchmarking numerical solutions by comparing them with analytical results. These limits also have clear physical relevance: waves with harmonic characteristics are often observed in narrow meridional domains, such as lakes (e.g., Simons, 1978), whereas waves with trapped characteristics appear in wide domains, for example in the Indian Ocean (De-Leon and Paldor, 2017). Comparing these basic wave theories with simple, idealized simulations provides a necessary foundation for studying wave dynamics in more complex numerical setups or in real-world circumstances.

Solving the inhomogeneous eigenvalue problem is interesting. But when you do this you are essentially decomposing the inhomogeneous part of the equation in terms of your eigenfunctions. These don't change at all and don't change the eigenfunctions. This is not something that we see very much in the literature and is worthy of further discussion.

We thank the referee for raising this point. We agree that it is useful to emphasize how the eigenfunction decomposition naturally arises in the solution of the initial-value problem. To address this, we have added a short discussion in Subsection 2.5 [following Eq. (14)]. There, we explicitly state that the general solution of the homogeneous problem can be expressed as a superposition of the eigenfunctions of Eq. (15), which ensures that any initial condition can be represented consistently using this basis. This added note clarifies the role of the eigenvalue problem in linking the initial-value formulation with the spectral solutions.

In section 5, when introducing the MITgcm, be explicit as to what equations you are solving. It does not solve the shallow water equations you have focused on up to this point.

We note that the MITgcm was configured to solve the same linear shallow water equations that are the focus of this study. The setup closely follows the procedure described in Section 4.1.1 ("Equations Solved") of the MITgcm barotropic gyre example (see `https://mitgcm.readthedocs.io/en/latest/examples/barotropic_gyre/barotropic_gyre.html`). Additionally, we removed the nonlinear terms in the material derivative and viscous dissipation from the MITgcm, leaving only the linear shallow water dynamics. This ensures that the MITgcm simulations are directly comparable to the analytical solutions of the linearized RSWE. Following the referee's comment, this clarification was explicitly added in the revised version of the manuscript (see P10).

In section 5, you now start to consider averages of the numerical solutions. But the theory does not mention temporal averages at all. If you want to have a good comparison, you should consider temporal averages in the linear theory.

The wave theories presented in the manuscript focus solely on the time-dependent component, $v'$, which is governed by the homogeneous part of Eq. (12), i.e., Eq. (14). In contrast, the MITgcm solves the RSWE, and therefore the simulations also include the time-independent, mean component, $\bar{v}(y)$, which corresponds to the solution of the inhomogeneous part of Eq. (12), i.e., Eq. (13). The comparison between the numerical simulations and the analytical wave solutions is carried out by subtracting $\bar{v}$ from the total velocity $v(y, t)$ of the numerical simulations as an estimate of the time-dependent component

$v'$. To explicitly clarify this point, we have revised the 1$^{\text{st}}$ paragraph on P11.

Page 11, you say the number of modes summed is $10^4$, then later its reduced to 500. How sensitive is your result to this number?

In the revised version, we clarify that in the calculation of the trapped wave solutions, the number of modes summed ($N$) was set to $10^4$, while for the harmonic wave solutions only, $N$ was reduced to 500. Multiplying $N$ by 2 or 0.5 has a negligible effect on the results. Figs. 1–2 below show $v'$ (for $L = 60$) in the geostrophic adjustment problem and the Ekman adjustment problem, respectively. In these figures, the harmonic wave solutions are shown by red curves, while the trapped wave solutions are shown by blue curves. The solid lines correspond to $N = 10^4$ (trapped waves) and $N = 500$ (harmonic waves). Dashed lines represent $N$ multiplied by 0.5, and dotted lines represent $N$ multiplied by 2. The overlap of the solid, dashed, and dotted curves in the figures clearly indicates that the insensitivity of the results to these changes in $N$. Reducing $N$ further to 50 has a more significant effect on the results (not shown).

Lots of emphasis is put on this epsilon parameter, the so called error estimator, for many different cases. I'm not convinced this is physically interesting. If you feel it is, please give some better justification for why you are considering it so intently.

We agree that the error measure $\epsilon(t)$, defined as the spatially averaged absolute difference between the theoretical and numerical fields, is not intended to represent a physically meaningful quantity on its own. Rather, its purpose is to provide a simple, systematic, and reproducible **quantitative** diagnostic that complements the visual comparisons presented in Figs. 1-4. The inclusion of $\epsilon(t)$ allows us to quantify the overall mismatch between theory and simulation in a way that is both concise and uniform for all cases. This helps us highlight the trends and systematic differences that might otherwise be overlooked in purely visual inspection.

To clarify this motivation, we revised the manuscript by adding a short subsection entitled *Error estimates* (Sec. 4.2), where the role of $\epsilon(t)$ is explicitly explained as a diagnostic tool rather than a physically interpretable parameter. We believe that this addition strengthens the justification for the use of epsilon as a single quantitative measure of accuracy. The entire discussion of the error measure in Sec. 4.2 is significantly shorter in the revised version.

Section 6 redoes everything for two layers. Why not do it all together? It was reduce what is a rather long manuscript.

We agree that the manuscript is rather long. Following the referee's comments, we moved the entire discussion of the two-layer and continuously stratified ocean models to the appendices so the discussion is now split into two appendices: Appendix C presents the analytical two-layer ocean model, while Appendix D describes the numerical simulations of a stratified ocean. Appendix C serves as an introduction and motivation for Appendix D.

Section 6.2 is called numerical simulations of a multilayered ocean and the authors use the MITgcm. This is not a layered model. Please be vary careful in your descriptions.

We thank the referee for the comment. To clarify, the MITgcm is not inherently a layered model. In our simulations, we employ a 3D configuration with 38 vertical grid cells (numerical layers) to represent a simplified two-layer physical ocean. The upper and lower physical layers correspond to groups of numerical layers, allowing a meaningful comparison with the two-layer analytical model. To make this distinction explicit, we have revised the opening paragraph of Appendix D to clearly emphasize the difference between physical layers (representing the conceptual two-layer ocean) and numerical layers (used for the model's vertical resolution).

In general, this paper seems to be a lot of different things thrown together and they don't necessarily flow very cleanly. Please work harder to make this a coherent story, not just a series of interesting results.

We greatly appreciated the referee's comment regarding the manuscript's coherence. In response, we have substantially shortened the main text and reorganized the material to present a more coherent narrative. The main text now focuses on the core problems, the analytical solutions, and comparisons with idealized single-layer simulations, while all technical derivations, the two-layer model analysis, multilayered simulations, and connections to observations have been moved to the appendices. This restructuring ensures that the manuscript now reads as a unified story, with the appendices providing all necessary details without disrupting the flow of the main discussion.

[Figure]

Figure 1: Geostrophic adjustment for $L = 60$. Red: harmonic wave solutions; blue: trapped wave solutions. Solid lines: $N = 500$ (harmonic) and $N = 10^4$ (trapped); dashed/dotted lines: $N$ multiplied by 0.5 and 2, respectively. Overlapping curves show negligible sensitivity to $N$.

[Figure]

Figure 2: Ekman adjustment for $L = 60$, using the same color and line conventions as Fig. 1. Curves overlap, indicating minimal sensitivity to variations in $N$.

---

## Author Response (AR2)

Author responses to Final editorial comments on second version of EGUSPHERE-2025-2529

Additional private note (visible to authors and reviewers only):

Editor comments are note in black and the lines in in the text where the comments were implemented are note in red

\* I would like that some of the content in the reply to referee 2 regarding dropping the b^2 y^2 term to feature in the article itself. The reply is ok, but none of that shows up currently in the article.

**L307-308; L185-189**

\* Above Eq (19): define what \xi and Ai are for completeness and avoid making the reader jump all the way to the Appendix for the definition. Could motivate a bit why it is the Airy function that shows up in the trapped wave scenario.

**L216-L217**

\* Section 4 paragraph 3: Be clear that it is "The MODIFIED MITgcm" that is meant, because as mentioned by referee 2 MITgcm is not a shallow water model. (And to be honest I thing using MITgcm is completely overkill for the present purpose; it's linear RSW so could just write bespoke code for it, or use something like Daedalus for example...)

**L264**

\* Fig 1 and elsewhere in the article: please be clear that in those cases t is units of rotational time  $f_0^{-1}$  (I assume). \Delta t in Table 1 currently is units of seconds so there was some ambiguity there.

**L291-2**

\* Not a requirement as such, but do acknowledgement referee(s) and/or editor if the authors think their contribution was actually helpful.

**Done**